# Prediction of Hardenability Curves for Non-Boron Steels via a Combined Machine Learning Model

**DOI:** 10.3390/ma15093127

**Published:** 2022-04-26

**Authors:** Xiaoxiao Geng, Shuize Wang, Asad Ullah, Guilin Wu, Hao Wang

**Affiliations:** 1Beijing Advanced Innovation Center for Materials Genome Engineering, University of Science and Technology Beijing, Beijing 100083, China; gengxiaoxiao1104@163.com (X.G.); guilinwu@ustb.edu.cn (G.W.); 2Department of Mathematical Sciences, Karakoram International University, Gilgit 15100, Pakistan; dr.asadullah@kiu.edu.pk; 3Yangjiang Branch, Guangdong Laboratory for Materials Science and Technology (Yangjiang Advanced Alloys Laboratory), Yangjiang 529500, China; 4School of Materials Science and Engineering, University of Science and Technology Beijing, Beijing 100083, China

**Keywords:** hardenability, machine learning, JMatPro, empirical formulas

## Abstract

Hardenability is one of the most basic criteria influencing the formulation of the heat treatment process and steel selection. Therefore, it is of great engineering value to calculate the hardenability curves rapidly and accurately without resorting to any laborious and costly experiments. However, generating a high-precision computational model for steels with different hardenability remains a challenge. In this study, a combined machine learning (CML) model including k-nearest neighbor and random forest is established to predict the hardenability curves of non-boron steels solely on the basis of chemical compositions: (i) random forest is first applied to classify steel into low- and high-hardenability steel; (ii) k-nearest neighbor and random forest models are then developed to predict the hardenability of low- and high-hardenability steel. Model validation is carried out by calculating and comparing the hardenability curves of five steels using different models. The results reveal that the CML model works well for its distinguished prediction performance with precise classification accuracy (100%), high correlation coefficient (≥0.981), and low mean absolute errors (≤3.6 HRC) and root-mean-square errors (≤3.9 HRC); it performs better than JMatPro and empirical formulas including the ideal critical diameter method and modified nonlinear equation. Therefore, this study demonstrates that the CML model combining material informatics and data-driven machine learning can rapidly and efficiently predict the hardenability curves of non-boron steel, with high prediction accuracy and a wide application range. It can guide process design and machine part selection, reducing the cost of trial and error and accelerating the development of new materials.

## 1. Introduction

Hardenability is the ability of steel to obtain martensite during quenching, depending on the austenization conditions and cooling rate [1]. Through the degree of hardenability, the most appropriate cooling medium is adopted to obtain the maximum depth of hardened layer under the condition of the minimum deformation of the workpiece. Steel has different applications depending on its hardenability. Low-hardenability steels have high surface hardness and good core toughness, and they are widely used in automobile gears, machine tool spindles, and industrial sectors [2]. High-hardenability steels normally have large quenchable section sizes of workpieces. Strong workpieces, i.e., springs, are usually made of high-hardenability steel to ensure that they can be quenched to a martensite structure [3]. Tools are also made of high-hardenability steels in order to obtain high strength, hardness, and wear resistance [4]. Therefore, hardenability can not only guide the design of the heat treatment process, but also serve as an important reference for the selection of machine parts [5].

The hardenability of steel is usually expressed as a hardenability curve. It is very laborious and costly to obtain hardenability curves through experiments, especially in a trial-and-error design. Therefore, calculation methods are used more often. As the hardenability mechanism of boron is different from other alloying elements, the calculation of hardenability of boron steel is complicated [6]. Trace amounts of boron (0.001–0.003 wt.%) can significantly improve the hardenability of steel. When the cooling rate is high, boron is adsorbed on the grain boundaries in an atomic state, which can inhibit and delay ferrite transformation and, thus, improve hardenability [7]. Upon decreasing the cooling rate, boron can be precipitated as boron phase, which leads to nonspontaneous nucleation and a sharp decrease in hardenability [8]. Therefore, only the calculation of hardenability curves of non-boron steels is discussed in this study. Empirical formulas based on experimental data and mathematical models based on phase transition dynamics were mainly used in the past to calculate hardenability curves. In recent years, machine learning models have been used to predict hardenability curves in order to ascertain the quenching behavior of steels. Grossman et al. [9] proposed the ideal critical diameter ($D_{I}$) method, which calculates $D_{I}$ on the basis of the chemical composition and grain size of steel, and then established the relationship between $D_{I}$ and the end-quenching curve. However, the calculation error was large, and it was only suitable for medium- and low-hardenability steel, hindering its application in practical production [10]. Just et al. [11] established regression equations to calculate the end-quenching curve on the basis of experimental data. However, their universality, accuracy, and rationality were also greatly limited, mainly because the linear mathematical model did not adapt to the actual shape and variation trend of the curve. Yu et al. [12] established nonlinear equations to predict the hardenability and mechanical properties. The modified nonlinear equation (MNE) was obtained through improvement, which further improved the prediction accuracy. Kirkaldy et al. [13] and Honeycombe et al. [14] calculated the hardenability of steel through the continuous cooling transformation (CCT) diagram of Jominy end-quenched bars on the basis of phase transformation kinetics and thermodynamics, and this method was integrated into the commercial software JMatPro. According to this model, the relationship between cooling rate temperatures $T_{cool\;rate}$ and end-quenching distances ϕ can be expressed as
(1)$$T_{cool\;rate}=-\left(T_{a}-297\right)\frac{4\eta }{\sqrt{\pi }xX^{2}}\phi^{3}\exp\left(-\phi^{2}\right),$$
(2)$$\phi =\sqrt{\frac{\pi }{2}}\left(\frac{T-297}{T_{a}-297}\right)+0.4406{\left[\frac{T-297}{T_{a}-297}\right]}^{3.725},$$
where *η* is the diffusivity at distance *X* (in cm) along the Jominy bar, *T* is the temperature, and *T_a_* is the austenization temperature. Thus, the new phase volume fraction *τ* can be expressed as Equation (3) to calculate the TTT diagram; then, Scheil’s addition rule [15] can be used to convert the TTT diagram to a CCT diagram. Each cooling rate of the CCT diagram corresponds to a data point of the Jominy end-quenching bar.
(3)$$\tau_{TTT}=\frac{1}{2^{\left(N/8\right)}{\left(\Delta T\right)}^{3}}e^{\left(\frac{Q_{eff}}{RT}\right)}\overset{m}{\underset{j=1}{\sum }}\alpha_{j}C_{j},$$
where α*_j_* is the constant for each element, *C_j_* is the concentration of the element, and *Q_eff_* is the effective activation energy for diffusion. The phase transformation reaction rate at any time is the derivative of the phase transformation volume fraction *τ* with time. Kirkaldy et al. believed that the inflection point of the phase transformation reaction rate is the same as that in TTT diagram and in proportion with the CCT diagram. Therefore, the points corresponding to the CCT diagram must correspond to the inflection point of the Jominy curve. Lastly, the corresponding Vickers pyramid number (VPN) is calculated as
(4)$$\begin{array}{c} VPN=Y1-\frac{\left(Y1-Y2\right)}{3X_{0}^{2}}X^{2},\;X<X_{0} \end{array},$$
(5)$$\begin{array}{c} VPN=Y2+\frac{2}{3}\left(Y1-Y2\right)\frac{X_{0}}{X},\;X>=X_{0} \end{array},$$
where *Y*1 and *Y*2 are the calculated hardness values of martensite and pearlite, respectively, in the alloy, and *X*_0_ is the distance from the inflection point of the Jominy curve to the quenching end.

With the development of the Materials Genome Initiative (MGI) [16] and Integrated Computational Materials Engineering (ICME) [17], data-driven machine learning methods have been gradually introduced into material design and development, with great achievements in various fields [18,19,20,21]. These methods can make rapid predictions on the basis of existing experimental data and effectively deal with the complex multivariate nonlinear relationship between input and output variables. Churyumov et al. [22] constructed an artificial neural network model for predicting flow stress of high-alloyed, corrosion-resistant steel during hot deformation. Honysz et al. [23] used generalized regression neural networks (GRNNs) to more accurately predict the chemical concentration of carbon and nine of the other most common alloying elements in ferritic stainless steels on the basis of the mechanical properties with the best efficiency. Artificial neural network algorithms have also been used to predict the hardenability of steel [24,25,26]. Gao et al. [27] applied polynomial regression and artificial neural networks to predict the hardenability of gear steels. Dong et al. [28] built a chemical composition–hardenability model for wear-resistant steels using an artificial neural network. However, these studies were modeled for specific steel grades; hence, the applicability of the models is limited. The change trends of high- and low-hardenability steel are different. Therefore, it is very necessary to obtain a calculation model with high accuracy and a wide application range.

In this study, a combined machine learning (CML) model is developed to effectively predict the hardenability of non-boron steels. Through collecting data, modeling, and evaluating classification and regression models, the optimal CML model is established. Due to the complex influence of composition, the hardenability curves of steels vary greatly. In order to improve the prediction accuracy and application range of the model, random forest (RF) is firstly used to divide the steel into low- and high-hardenability steel. Then, k-nearest neighbor (k-NN) is used to predict the quenching curve of low-hardenability steel, while random forest is used as the optimal prediction model for high-hardenability steel. To further verify the accuracy of the CML model, the hardenability curves of five steels are predicted by this model and other methods. The results show that the hardenability curve calculated by CML model is in excellent agreement with the experimental ones, and the prediction accuracy of this model is better than that of JMatPro software, the ideal critical diameter method, and the modified nonlinear equation. Therefore, the CML model combining material informatics and data-driven machine learning can quickly and efficiently predict the hardenability curve of non-boron steel, with high prediction accuracy and a wide application range. It has certain guiding significance for heat treatment process design and mechanical part selection, which can accelerate new material research and development.

## 2. Methodology

### 2.1. Data Collection and Preprocessing

In this study, the hardenability curves of 126 different steels were obtained from published studies [2,29,30,31], of which 121 groups were used for training and five groups were used for testing. The ranges of composition, austenitizing temperature (AT), and Jominy equivalent cooling rate ($J_{ec}$) are shown in Appendix A. Due to the complex influence of alloying elements, the hardness variation trends of low- and high-hardenability steels are different. The hardenability curves of low-hardenability steel (5SiMnMoV) and high-hardenability steel (40CrNiMoA) are shown in Figure 1. In the figure, the horizontal axis is the distance to the quenching end X (mm), while the vertical axis is the hardness (HRC). It can be seen that the hardness of low-hardenability steel decreases continuously from the water-cooled end until it approaches the horizontal line. Due to the low critical cooling rate of high-hardenability steel, martensite can be obtained in a range greater than the critical cooling rate. Therefore, its hardness does not change much within a certain distance from the water-cooled end, beyond which it begins to decline. The hardenability curves or hardenability bands collected in this study were obtained from end-quenched specimens using the Jominy method. For certain grades of steel, because their chemical composition fluctuates within a certain range, hardenability bands are obtained rather than a line. In order to obtain more statistically significant data, the average value of the upper and lower limits of the hardenability bands should be taken as the hardenability curve. These hardenability curves can be converted from graphical format to numerical format, i.e., distance versus hardness (X-HRC). All points were taken manually, and hardness values were obtained at 1.5 mm or 3 mm intervals along the horizontal axis, resulting in a training set and test set. Then, the training and test datasets were converted to CSV file format. Each dataset contained 11 input variables, whose attributes were numeric. The number of instances in the training datasets of low- and high-hardenability steels is shown in Appendix A.

### 2.2. Feature Selection

Feature selection is the key step in the data analysis process, which greatly affects the results of machine learning. Chemical composition and austenite grain size are some of most important factors affecting hardenability [32]. However, austenite grain size was not used as a feature parameter due to a lack of relevant data. Therefore, the input feature parameters of the present model included the chemical composition (C, Si, Mn, Cr, Ni, Mo, W, V, Ti, Cu) and the distance X along the Jominy bar (1.5, 3, 6, 9, 12, 15, 18, 21, 24, 27, and 30 mm). The output feature parameter was the hardness value (in HRC). The relationship between the hardness value and the selected features can be described by Equation (6).
(6)$$HRC=f\left(C,Si,Mn,Cr,Ni,Mo,W,V,Ti,Cu,X\right).$$

In order to further verify the relationship between alloying elements and hardenability, a Pearson correlation map was calculated using the training data with blue and red colors indicating positive and negative correlations, respectively. A lighter tone indicates a less significant corresponding correlation. The filled fraction of each pie chart in the graph corresponds to the absolute value of the associated Pearson correlation coefficient. Here, the hardness 15 mm away from the cold quenching end was taken as the hardenability index, marked as J15. The Pearson correlation map for J15 and chemical elements is shown in Figure 2. It can be seen that C had the greatest influence on hardenability of steels, while hardenability increased with increasing C content. Alloying elements Cr, Ni, Mo, W, and Si had positive correlations with hardenability, indicating that they could improve hardenability to varying degrees. Micro-alloy elements V and Ti were negatively correlated with hardenability, indicating that their addition reduced hardenability. For example, V can consume C in the solution to form carbonitride, thus reducing hardenability [33]. Ti probably reduces hardenability for the same reasons. The correlation of Mn and Cu with hardenability was not obvious, indicating that Mn and Cu were not linearly correlated with hardenability. In summary, the influence of alloying elements on hardenability was basically consistent with the common knowledge of materials.

### 2.3. Machine Learning Models

A variety of machine learning algorithms were applied to build prediction models of hardenability curves using Weka 3.9. K-NN is a nonparametric method used for classification and regression [34]. The mean of top k labels is used for the regression task, while the mode of top k labels is used in the case of classification. Multilayer perceptron is a feedforward artificial neural network model that is trained on a set of input–output pairs and learns to model the correlation between those inputs and outputs. Training involves adjusting the parameters or weights and biases of the model in order to minimize error. Backpropagation is used to make these weight and bias adjustments relative to the error. Multilayer perceptron classifier (MLP-C) and multilayer perceptron regression (MLP-R) can handle classification and regression problems, respectively, and MLP-R was mostly used to predict the hardenability curves in previous studies [24,25,26]. SVC is a classifier that is used for predicting discrete categorical labels, while SVR is a regression algorithm that supports both linear and nonlinear regressions, used for predicting continuous ordered variables [35]. Radial basis function (RBF) networks were trained in a fully supervised manner using WEKA’s optimization class by minimizing the squared error according to the BFGS method [36]. Bagging (bootstrap aggregating) is a simple and effective integration method to obtain training subsets on the basis of uniform random sampling with replacement [37]. RF is an ensemble learning method for classification, regression, and other tasks that operates by constructing a multitude of decision trees at training time, which can handle complex multivariable nonlinear problems with good generalization ability and resistance to overfitting [38].

### 2.4. Model Evaluation

In order to reduce the influence of unfitting and overfitting problems on calculations, appropriate evaluation methods and indicators need to be chosen for established models. In this paper, 10-fold cross-validation was adopted, whereby the original sample set was randomly divided into 10 equal-sized subsample sets, among which nine subsample sets were used as training data and the remaining subsample set was used as validation data [39]. By repeating the above process 10 times, each subsample set was used as validation data only once. For the classification model, the evaluation parameters were the values of accuracy [40], F1-score [41], and area under curve (AUC) [42]. Accuracy can reflect the ability of a classification model to judge the whole sample, as shown in Equation (7), where TP is true positive, FP is false positive, FN is false negative, and TN is true negative. The F1-score can be regarded as the weighted average of accuracy and recall rate, as shown in Equation (8). Higher values of accuracy and F1-score indicate better discriminability of the model. AUC is the area under the receiver operating characteristic (ROC) curve, which enables a reasonable evaluation of the classifier in the case of unbalanced samples. The classifier performs better if the AUC value is closer to 1.
(7)$$Accuracy=\frac{TP+TN}{TP+FP+FN+TN}.$$
(8)$$F1-Score=2\cdot \frac{\mathrm{precision}\cdot \mathrm{recall}}{\mathrm{precision}+\mathrm{recall}\;}.$$

One of the evaluation indicators of a regression model is the correlation coefficient (CC, Equation (9)) [43], where $x_{i}$, $y_{i}$ are the experimental values and predicted values, and $\bar{x}$, $\bar{y}$ are the corresponding average values, respectively. CC is a real number between −1 and 1, and a greater absolute value represents a higher correlation between input and output features. Generally, an absolute value of CC greater than 0.8 is considered as highly correlated. In this study, the minimum CC standard was set to be 0.95 to ensure reliable prediction results.
(9)$$\begin{array}{c} CC_{xy}=\frac{\sum \left(x_{i}-\bar{x}\right)\left(y_{i}-\bar{y}\right)}{\sqrt{\sum {\left(x_{i}-\bar{x}\right)}^{2}\sum {\left(y_{i}-\bar{y}\right)}^{2}}}. \end{array}$$

The mean absolute error (MAE, Equation (10)) and root-mean-square error (RMSE, Equation (11)) are used to evaluate the difference between experimental and predicted values [43], where $x_{i}$ is the experimental value, $y_{i}$ is the predicted value, and *n* is the number of samples. Lower values of MAE and RMSE denote better consistency between measured and predicted hardness, as well as a more accurate learning model. According to the evaluation thresholds of CC > 0.95 and MAE/RMSE < 3 HRC, an optimal model was obtained to predict hardenability.
(10)$$\begin{array}{c} MAE\left(y,\hat{y}\right)=\frac{1}{n}\;\overset{n}{\underset{i=1}{\sum }}\left|x_{i}-y_{i}\right|. \end{array}$$
(11)$$\begin{array}{c} RMSE\left(y,\hat{y}\right)={\left[\frac{1}{n}\;\overset{n}{\underset{i=1}{\sum }}{\left|x_{i}-y_{i}\right|}^{2}\right]}^{\frac{1}{2}} \end{array}.$$

## 3. Results and Discussion

### 3.1. Classification Model

The mutual interaction of various alloying elements in steel is undoubtedly complex, which leads to a significant difference in the hardenability curves of steels. Therefore, the preliminary classification of steels based on the calculated $D_{I}$ was the first stage of the hardenability modeling method. The calculation formula of $D_{I}$ is shown in Equation (12).
(12)$$D_{I}=25.4\times f_{GZ}\cdot f_{C}\cdot f_{Mn}\cdot f_{Si}\cdot f_{Ni}\cdot f_{Cr}\cdot f_{Mo}\cdot f_{Cu}\cdot f_{V},$$
where $f_{GZ}$ is the calculation factor of the grain size grade, and $f_{C}$, $f_{Mn}$, $f_{Si}$, $f_{Ni}$, $f_{Cr}$, $f_{Mo}$, $f_{Cu}$, and $f_{V}$ are the calculation factors of alloying elements, which can be obtained from the American Society for Testing Materials A255-20a (ASTM A255-20a). In this study, $f_{GZ}$ was 1.0, corresponding to the calculated factor value for a granularity of 7. When the calculated $D_{I}$ of steel was greater than 80, it was regarded as a high-hardenability steel. Otherwise, it was considered as a low-hardenability steel.

Classification models were established using bagging, k-NN, MLP-C, SVC, and RF. Figure 3 shows the training results. As can be seen, the accuracy rate and F1-score value of RF were the highest, both reaching 92.1%. The AUC of RF was also the maximum, being 0.953. The accuracy and F1-score of k-NN and MLP-C were 91.3%. However, the AUC of k-NN was 0.917, lower than that of RF. MLP-C had a similar AUC value to RF, but RF had higher accuracy and stronger generalization ability than MLP-C. Therefore, RF was taken as the optimal classification model.

### 3.2. Regression Model

Regression models included the bagging, k-NN, MLP-R, SVR, RBF, and RF models. In order to make a more intuitive comparison, scatter diagrams were adopted to show the training results of these models, as shown in Figure 4 and Figure 5. In these scatter plots, darker colors represent smaller absolute values in error. For absolute values of error greater than 18 HRC, the point was colored gray. More points concentrated on the line *y* = *x* indicated better performance of the model.

For low-hardenability steels, as shown in Figure 4, the predicted values of k-NN were more concentrated on the oblique 45° line compared with other algorithms; k-NN had the highest trained CC value of 0.983 and the smallest MAE and RMSE, 1.2 HRC and 2.2 HRC, respectively. On the other hand, SVR had the worst prediction performance with CC equal to 0.773, and MAE and RMSE equal to 5.9 and 7.5 HRC, respectively. It can also be seen that the prediction values of k-NN had the strongest correlation with the experimental data.

Figure 5 shows the results of the training model for high-hardenability steels. It can be seen that the calculated data points were more concentrated in the vicinity of oblique 45° reading line for RF, with the largest CC value (0.994) and the smallest error values, with MAE being 0.75 HRC and RMSE being 1.1 HRC. The training results of SVR were also the worst, with the lowest CC value and the largest error values, indicating that SVR is not suitable for the regression prediction of the hardenability of steels. Since the RF algorithm performed very well for both high-hardenability steels and low-hardenability steels, it was further used to establish the optimal prediction model of the hardenability of steels.

### 3.3. Model Validation

In order to verify the accuracy of the classification model and further determine the optimal regression model, the hardenability curves of five randomly selected steels with different compositions were chosen as test samples. The chemical compositions of these test samples are shown in Appendix A. The number of instances in the test datasets is shown in Appendix A. Among these steels, #1 and #2 steels were low-hardenability steels, while #3, #4, and #5 steels were high-hardenability steels. The hardenability curves of these five steels were used as the verification set; thus, data of these five steels were not included in the training set. RF could accurately classify steels, playing a very important role in the subsequent calculation of the regression model. These five steels were classified by RF, and an accuracy of classification of 100% was obtained, indicating that RF could accurately classify steels, enabling the hardenability curve to be accurately calculated by the regression model.

Six regression models were then used to predict the hardenability curves of steels. The correlation coefficients and error values are shown in Figure 6. For #1 steel, the CC value of k-NN was relatively higher than others; the MAE and RMSE of k-NN were the smallest, 2.2 HRC and 3.0 HRC respectively. RF was better than k-NN in predicting the hardenability curve of #2 steel. However, the performance of RF was worse than that of k-NN in the training set. Therefore, k-NN was chosen as the optimal prediction model for low-hardenability steels. For high-hardenability steels, it can be seen that the CC values of all algorithms were all higher than 0.95. For #4 and #5 steels, the error values of RF were the smallest, with MAE being 0.3 and 0.4 HRC and RMSE being 0.8 and 0.8 HRC, respectively. As for #3 steel, the data distribution led to SVR having the smallest MAE and RMSE in the test set. However, the MAE and RMSE of SVR for #4 and #5 steel were higher than those of RF, and the performance of SVR in the training set was the worst. Therefore, SVR was not considered as the optimal model. RF was, thus, selected as the prediction model for high-hardenability steel.

Finally, these classification and regression models were combined to form a CML model to calculate the hardenability curves of non-boron steels, and the flow chart is shown in Figure 7. In the CML model, non-boron steels were divided into high-hardenability steels and low-hardenability steels by RF. Then, the k-NN model was established to predict the hardenability curves of low-hardenability steels, and the RF model was used for the calculation of the hardenability curves of high-hardenability steels. The optimized parameters and package version of the models are shown in Appendix A. The applicability of the CML model depends largely on the training dataset and the generalization ability of the arithmetic. Therefore, it can be considered that the present CML model has relatively high prediction accuracy for steel in the range of compositions listed in Appendix A.

### 3.4. Comparison of CML Model with Others

In order to further verify the accuracy of the present CML model, the prediction results of CML were compared with the $D_{I}$ method, the MNE model, and the commercial software JMatPro. In the empirical formula and JMatPro, the grain size was also set to 7. The comparison of errors calculated by different methods is shown in Figure 8, and the comparison results of calculated hardenability curves are shown in Figure 9, where the black line indicates the experimental results, while the red, blue, orange, and green lines represent the prediction results of the CML, JMatPro, $D_{I}$, and MNE models, respectively. It can be seen that the present CML model could accurately predict the hardenability curve of steels and was superior to the $D_{I}$ method, the MNE model, and JMatPro. The present CML model had the smallest calculation errors, and the predicted hardenability curves of CML were more consistent with the experimental ones than the curves of others, especially for the #1, #4, and #5 steels. For example, the MAE and RMSE of CML were both smaller than 3 HRC, except for #2 steel, but the errors of #2 steel were still much smaller than those of other models. For #1 and #2 steels, the overall change trends predicted by JMatPro were away from experimental ones. The calculation error of JMatPro started to increase when X was greater than 10 mm for #3 and #4 steels. Although the MAE and RMSE of JMatPro for #5 steel were small, 1.4 HRC and 1.6 HRC, respectively, the hardenability curve predicted by the CML model was closer to the experimental one than that of JMatPro. Therefore, the accuracy of JMatPro for the overall prediction of the steel hardenability curve was not high enough. Compared with CML model, the errors of MNE and $D_{I}$ were quite large for both low-hardenability steels and high-hardenability steels.

In summary, an effective CML model to predict hardenability curves of steels was established only on the basis of chemical compositions and distance along the Jominy bar, making the prediction of hardenability curves more convenient and accurate during alloy design.

## 4. Conclusions

A combined machine learning (CML) model including classification and regression was developed to predict the hardenability curves of non-boron steels using chemical composition and distance along the Jominy bar. Bagging, k-NN, MLP-C, MLP-R, SVC, SVR, RBF, and RF algorithms were applied on experimental datasets, and an optimal model was selected by comparing the results of 10-fold cross-validation, in terms of correlation coefficients and error values. In the present CML model, steels are first classified into low-hardenability or high-hardenability steel by RF. For low-hardenability steels, k-NN had the best prediction performance and was, thus, selected as the optimal model. For high permeability steels, RF was adopted as the optimal model for hardenability curves.

The accuracy of the present CML model was verified by comparing the predicted results with experimental ones and prediction results of JMatPro, the ideal critical diameter method, and modified nonlinear equations. A superior predictive performance was obtained by the present CML model with a classification accuracy of 100%, high correlation coefficients, and low error values.

The findings of this work can realize the potential of big data mining. By using existing experimental data and machine learning algorithms to calculate the process curves and mechanical properties of steels, the experimental cost can be reduced, and the development of new materials can be accelerated.

## Figures and Tables

**Figure 1 materials-15-03127-f001:**
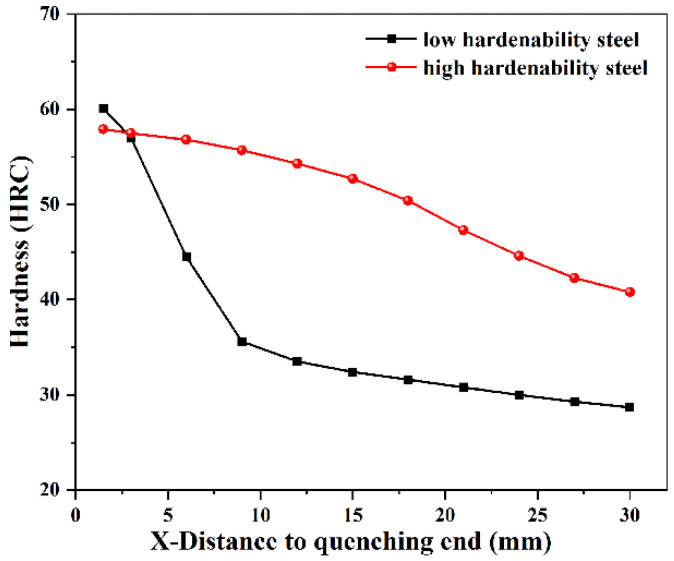
Hardenability curves for low- and high-hardenability steels.

**Figure 2 materials-15-03127-f002:**
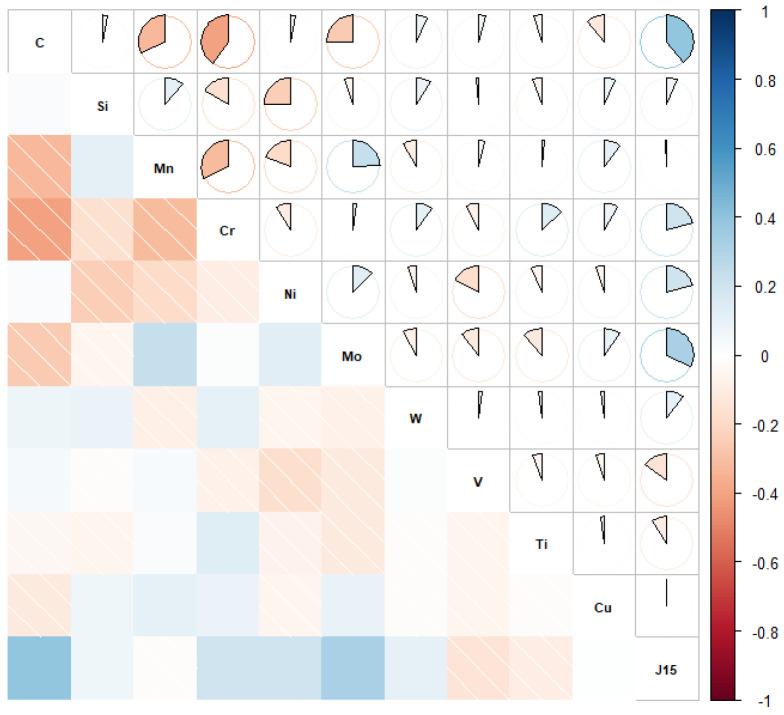
Pearson correlation map of hardenability and chemical elements.

**Figure 3 materials-15-03127-f003:**
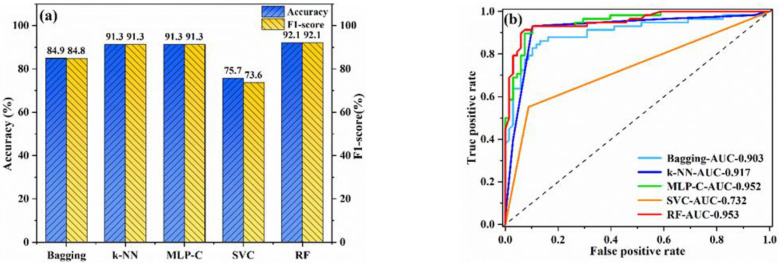
The performance of classification models on the training set: (**a**) accuracy and F1-score; (**b**) ROC curves.

**Figure 4 materials-15-03127-f004:**
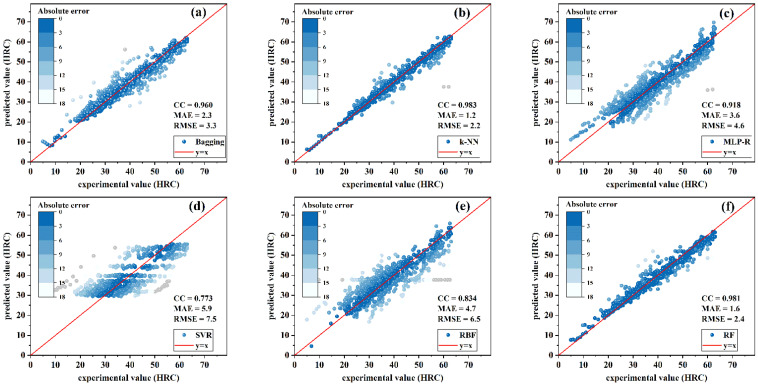
Training results of regression models for low-hardenability steels: (**a**) bagging; (**b**) k-NN; (**c**) MLP-R; (**d**) SVR; (**e**) RBF; (**f**) RF.

**Figure 5 materials-15-03127-f005:**
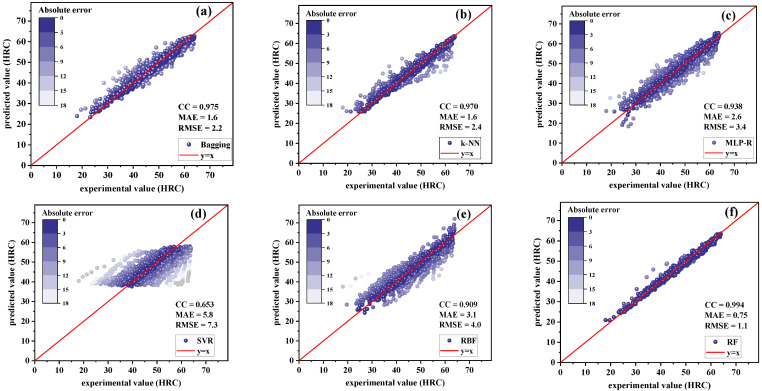
Training results of regression models for high-hardenability steels: (**a**) bagging; (**b**) k-NN; (**c**) MLP-R; (**d**) SVR; (**e**) RBF; (**f**) RF.

**Figure 6 materials-15-03127-f006:**
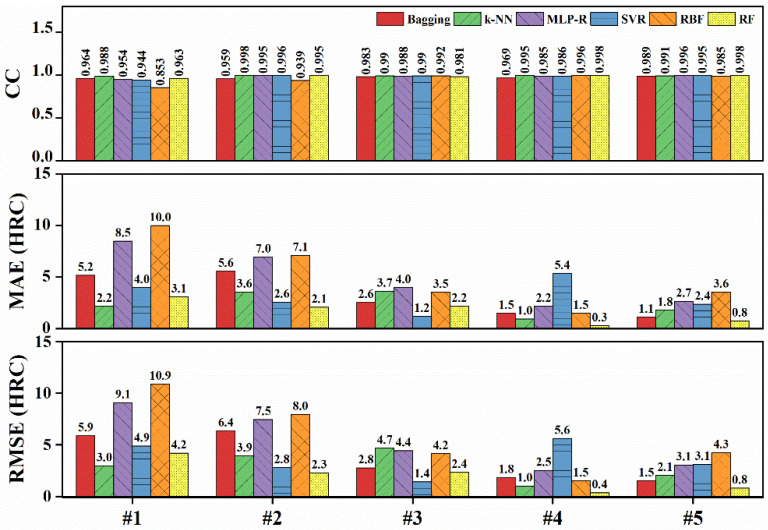
Correlation coefficients and error values of the prediction result on the test sets.

**Figure 7 materials-15-03127-f007:**
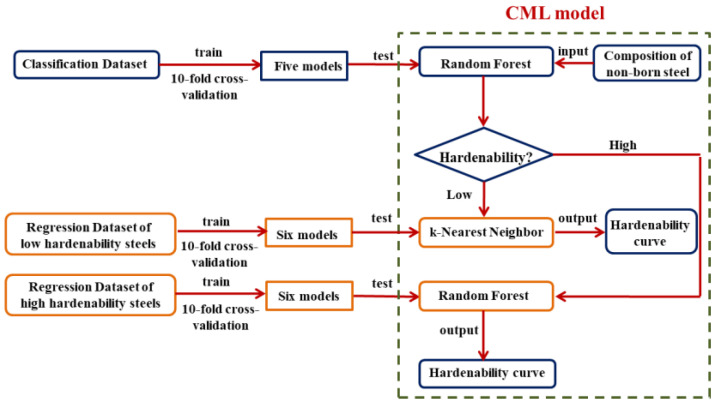
Flow chart of the CML model for predicting hardenability curves of non-boron steels.

**Figure 8 materials-15-03127-f008:**
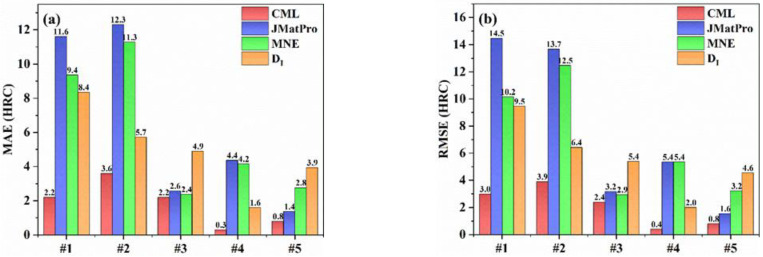
Comparison of the error values calculated by different calculation methods on the test set: (**a**) MAE; (**b**) RMSE.

**Figure 9 materials-15-03127-f009:**
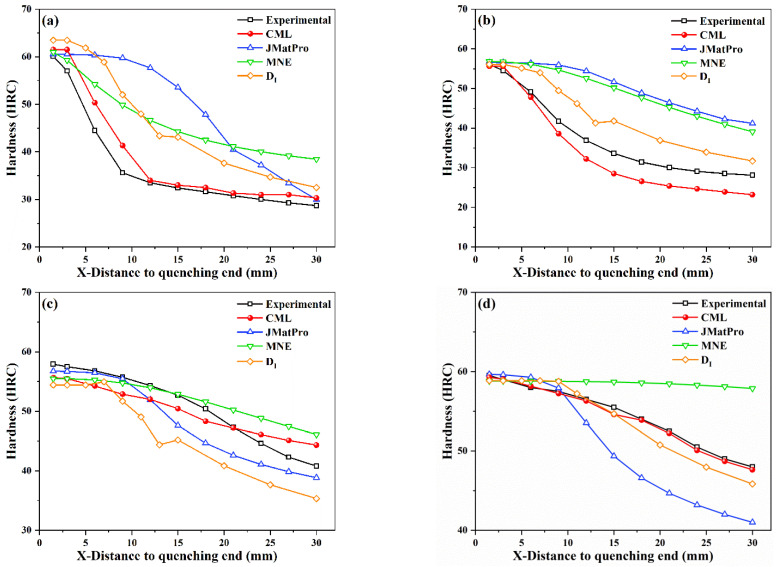
Comparison of experimental and predicted hardenability curves of steels by CML, JMatPro, MNE, and $D_{I}$; (**a**) #1-5SiMnMoV; (**b**) #2-42SiMn; (**c**) #3-40CrNiMoA; (**d**) #4-45CrMnMo; (**e**) #5-50CrMnVA.

## Data Availability

The data presented in this study are available on request from the corresponding author.

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
