# Peer review of "Prediction of Hardenability Curves for Non-Boron Steels via a Combined Machine Learning Model"

_materials, 2022, doi:10.3390/ma15093127_

Round 1
Reviewer 1 Report
The fast rise in the power of computers and large datasets of experimental data provide an opportunity to calculate the properties of materials without any experimental work. In the paper "Prediction of hardenability curves for non-boron steels via a combined machine learning model", the authors construct the combined machine learning model for the determination of the low-alloyed steels’ hardness after the quenching. The constructed model shows high accuracy and may be useful for the determination of new steels' hardenability. The presented results seem to be interesting. However, some parts of the paper are needed to be modified accordingly following comments:
- Why the authors have constructed the model only for the non-boron steels? Is that element has a principal meaning on the hardenability in comparison with other elements? Additional information should be added to the manuscript.
- Most of the references in the Introduction part are too old. It is recommended to analyze more new papers about modeling of the steels’ properties using the ANN approach (e.g., 10.3390/met12030447, 10.3390/met11050724, etc).
- It is recommended to add the table with the range of the elements in the steels that were used for the model’s construction such as the range of the quenching temperatures and cooling rates. It should be very useful for the readers to determine the limits of the model.
- In the Abstract and conclusion the authors give only the classification accuracy of 100%. It is better to change this parameter to the absolute average relative error (please, see the uploaded file).
- The authors have constructed the combined machine learning model for predicting the hardenability of the steel. But it is not clear how other readers will be able to use this model? It is recommended to provide the regression equation and coefficients as Supplementary files to the manuscript.

Author Response
- Why the authors have constructed the model only for the non-boron steels? Is that element has a principal meaning on the hardenability in comparison with other elements? Additional information should be added to the manuscript.
Response: We have added some comments related to the different hardenability mechanisms of boron in the introduction section. As the hardenability mechanism of boron is different from other alloying elements, the calculation of hardenability of boron steel is complicated. Trace amounts of boron (0.001~0.003 wt. %) can significantly improve the hardenability of steel. When the cooling rate is high, boron is adsorbed on grain boundaries in atomic state, which can inhibit and delay ferrite transformation and thus improve hardenability [1]. With decreasing cooling rate, boron can be precipitated as boron phase, which leads to non-spontaneous nucleation and a sharp decrease in hardenability [2]. Therefore, only the calculation of hardenability curve of non-boron steels is discussed in this study. The added contents are labeled in yellow color.
[1] Kamada Y, Kurayasu H, Watan AbES. Relation between hardenability and segregation to austenite grain boundaries of boron atom on direct quenching process. Tetsu to Hagane, 2010;74(11):2153-2160.
[2] Biaobrzeska B. Effect of alloying additives and microadditives on hardenability increase caused by action of boron. Metals. 2021; 11(4):589.
- Most of the references in the Introduction part are too old. It is recommended to analyze more new papers about modeling of the steels’ properties using the ANN approach (e.g., 10.3390/met12030447, 10.3390/met11050724, etc).
Response: We added 2 references and relevant contents to support for predicting steels’ properties using ANN methods.
- It is recommended to add the table with the range of the elements in the steels that were used for the model’s construction such as the range of the quenching temperatures and cooling rates. It should be very useful for the readers to determine the limits of the model.
Response: We added the range of austenitizing temperature (AT) and Jominy equivalent cooling rate ( ) in the supplementary table S-I. The hardenability curves collected in this study are obtained from end quenched specimens by Jominy test. In the Jominy test, the distance along the Jominy bar represents the cooling rate. In this study, the distance X along the Jominy bar are 1.5, 3, 6, 9, 12, 15, 18, 21, 24, 27 and 30 (in mm) respectively, as shown in the section 2.2. So we give the range of equivalent cooling rates for corresponding distances.
- In the Abstract and conclusion the authors give only the classification accuracy of 100%. It is better to change this parameter to the absolute average relative error (please, see the uploaded file).
Response: Generally speaking, the commonly used evaluation parameters for classification models are accuracy, F1-score and area under curve (AUC). The absolute average relative error is more suitable for regression model. In this study, the 121 steels were included in the training set, and the accuracy of the best classification model on the training set was 92.1%. The test set contains 5 steels with different components. The classification accuracy of the model in the test set of 5 steels is very precise, i.e. 100%, indicating that the 5 steels can be accurately divided into high and low hardenability steels.
- The authors have constructed the combined machine learning model for predicting the hardenability of the steel. But it is not clear how other readers will be able to use this model? It is recommended to provide the regression equation and coefficients as Supplementary files to the manuscript.
Response: We have shown the optimized parameters and package version of models in the Supplementary table S-III.

Reviewer 2 Report
What samples (dimensions) of materials were applied?
Equation (12) shows the notation of factors (fi). Another indication is in the text (page 230). It needs to be adjusted
In equations 9, 10, 11, match the designation of the factors with the designation in the text (eg arithmetic mean, calculated values) on page 214
Figures 4 and 5 show the results of low and high hardenability regression model tests. The set of points, respectively, corresponds to how many measurements?
The methodology is fine.
If the chemical composition is in the tables, according to the content of this article there are references to tables S-I (page 118), S-II (page 139), S-III (page 302), S-IV (page 274), S-V (page 275) but from what source were they used.
Based on that, I can't relevantly highlight the design model.
Author Response
- What samples (dimensions) of materials were applied?
Response: Materials: all kinds of alloyed steels containing no boron element. The sizes of test samples are defined by national standards GBT 225-2006. According to this standard, the dimensions are shown as Fig.1.
Fig.1. Size of test samples (a) flanged samples (b) fluted samples
- Equation (12) shows the notation of factors (fi). Another indication is in the text (line 230). It needs to be adjusted.
Response: Line 230 is the explanation of equation 12. So, these two indications are the same. We modified the language to make this point clear.
- In equations 9, 10, 11, match the designation of the factors with the designation in the text (eg arithmetic mean, calculated values) on line 214.
Response: According to the reviewer’s suggestion, we modified the language of line 214 to make the factors in equations 9, 10 and 11 matched the designation.
- Figures 4 and 5 show the results of low and high hardenability regression model tests. The set of points, respectively, corresponds to how many measurements?
Response: We have shown the number of instances in the training datasets of low and high hardenability in the supplementary table S-III. The training set for low and high hardenability steels contained 827 and 1032 measurements, respectively.
- If the chemical composition is in the tables, according to the content of this article there are references to tables S-I (page 118), S-II (page 139), S-III (page 302), S-IV (page 274), S-V (page 275) but from what source were they used.
Response: Table S-I contains the range of chemical composition, austenitizing temperature and Jominy equivalent cooling rate of steels in the training set, which can assist in better understanding of the application scope of the model. Table S-II shows the number of training datasets for low and high hardenability steels, which can help us judge the rationality of modeling. Table S-III shows the optimized parameters and package version of the algorithm, which is convenient for the application of the models. Table S-IV and table S-V show the chemical composition of five steels tested and the number of instances in each test set to better understand the predictive performance of the models on the test set. Table S-I and table S-II are from the training sets of the model, table S-III is from modeling parameters, and table S-IV and S-V are from the test sets of the model.
Round 2
Reviewer 1 Report
The authors have answered the previous comments and significantly improved the manuscript. The paper may be accepted in the current state.
Minor revision: Line 105: Honysz is a sole author of the paper [23] and the phrase "et al" is unnecessary.
Reviewer 2 Report
I have no further comments